# Decrease in Primary Caregivers’ Quality of Life During the Care of a Relative with Palliative Care Needs: A Prospective Longitudinal Study

**DOI:** 10.3390/cancers16213570

**Published:** 2024-10-23

**Authors:** Rodica Sorina Pop, Bianca Olivia Cojan Minzat, Cristina Paula Ursu, Ștefan Ursu, Aida Puia

**Affiliations:** 1Department Family Medicine, Faculty of Medicine, University of Medicine and Pharmacy “Iuliu Hatieganu”, 400337 Cluj Napoca, Romania; drsorinapop@yahoo.com (R.S.P.); cristinapaulapop10@yahoo.com (C.P.U.); stefan.ursu20@gmail.com (Ș.U.); draidapuia@gmail.com (A.P.); 2Family Medicine Office, 400259 Cluj-Napoca, Romania; 3Municipal Hospital “Dr Cornel Igna”, 405100 Campia Turzii, Romania; 4Municipal Hospital, 435700 Viseul de Sus, Romania; 5Regional Institute of Gastroenterology “Prof. Dr. Octavian Fodor”, 400394 Cluj Napoca, Romania

**Keywords:** caregivers, quality of life, patient, cancer, non-malignant

## Abstract

The quality of life is influenced by individual’s characteristics and values, personal’s beliefs, physical and mental health and the quality of relationship to others. The researchers want to investigate how the quality of life of primary caregivers changes during care for patient with palliative care needs. It is a prospective longitudinal study that assesses the different aspects of the quality of life of caregivers in care process of palliative care. The caregiver burden which increased during the care of patient have a great and negative impact on quality of life as decrease significantly physical, psycho-emotional and social aspects of this. The primary caregiver is feeling the perception of deteriorating general health.

## 1. Introduction

The quality of life is a complex concept without a standard definition. According to the World Health Organization (WHO), “quality of life” is influenced by physical health, psycho-emotional status, personal beliefs, social relationships, and the various characteristics of a person’s living environment [1]. There are five aspects of quality of life: normal life, satisfaction, achievement of personal goals, social utility, and functionality [2,3].

The primary caregiver is defined as the person who takes complex medical and nursing responsibilities in patient care. This person is identified by the patient and shares the patient’s entire experience with the illness [4]. Most of the time, the primary caregiver is a family member but sometimes it can be any other person who is directly and actively involved in the patient’s care without being paid [5]. For the health system, the primary caregivers represent an extension of professional staff at the patient’s home [6].

The quality of life of primary caregivers, who care for patients with cancer, is greatly decreased [7] by their multitude of various responsibilities [8] but especially by severe psycho-emotional tension [9,10] compared with that in the general population. Research has shown an increased mortality risk for primary caregivers [11].

However, many aspects that influence the patients’ and their family caregivers’ quality of life are unknown and inconstantly related [12]. The caregiver’s burden of the patient’s symptoms may vary in different diseases [13] and it is influenced by the illness’ trajectory [14], the type of treatment at the end-of-life stage [15], and the family caregivers’ stress and worries regarding care [16,17].

The primary caregiver’s assessment showed that his quality of life is impacted by the burden of care [17,18].

### Aim of Study

This prospective longitudinal study aims to understand how the quality of life of family members who take on the role of primary caregivers for a relative who is a palliative care patient is influenced by this role.

Research questions:How does the quality of life of the primary caregiver of a patient with palliative needs change?Are there differences between the group of cancer patients and the group of non-cancer patients in terms of the quality of life of the primary caregivers?

## 2. Materials and Methods

During a prospective longitudinal study, the quality of life of primary caregivers was measured and compared between two groups: primary caregivers of palliative patients with non-malignant diseases (PrC_1_) and primary caregivers of palliative patients with cancer (PrC_2_). A longitudinal study was chosen to investigate correlations between aspects of the primary caregiver’s quality of life over a period of three months.

### 2.1. Sample and Setting

Caregivers were recruited from the Palliative Care Department when they came with a patient for hospitalization or consultation. The patients’ primary caregivers, who were over 18 years old, were Romanian speakers, did not receive money for the care they provided, did not have a disease affecting their cognition, and gave their written agreement to participate in the study, were the subjects of this investigation. All the patients cared for by the study subjects met the criteria for inclusion in palliative care. The patient caregivers included in this study were divided into two groups: those who cared for cancer patients (PrC_2_) and those who cared for patients with non-malignant diseases (PrC_1_) (Figure 1).

### 2.2. Methods

Upon enrolment in the study, at the initial moment (T0), the PrC filled out the questionnaire with the following demographic data: age, sex, living environment, degree of kinship with the patient, occupation, level of education, place of residence, distance between the patient’s home and the caregiver’s home and the number of hours spent caring for the patient or for other family members. In the first meeting, the caregiver completed Medical Outcomes Scale-Short Form 36 (MOS-SF36) to measure their quality of life. This form was completed monthly for a period of three months (T1, T2, T3). The percentage of patients lost due to death was high during the study period. If the patient died during this period, the caregiver was assessed two months after the patient’s death (Tf).

The MOS-SF-36 is a short-form instrument for measuring quality of life in the general population. This tool is appropriate because the primary caregivers are part of the general population. This scale has 36 items that assess eight aspects of quality of life and health: physical functionality (PhF), limitations in one’s usual role due to physical health (LPh), limitations in one’s usual role due to emotional problems (LE), social functioning (SOC), bodily pain (P), perceptions about general health (GH), vitality (energy/fatigue), (V) and emotional well-being (EM) [19,20]. The score can be from “0” to “100” and a high score corresponds to a more favorable health state. This instrument has good internal consistency [21].

### 2.3. Statistical Analyses

Descriptive and analytic statistics were performed using IBS SPSS v26.0. Demographic characteristics were analyzed using simple statistical methods appropriate for the various types of variables. The Mann–Whitney U test was used to compare outcomes between two independent groups, PrC1 and PrC2. The Kruskal–Wallis H test extends the Mann–Whitney U test and is used to compare the differences between two or more groups of an independent variable, respectively, at T0, T1, T2, or T3. Distribution of the values was inhomogeneous.

## 3. Results

A total of 146 patients presented to the hospital during the research period but only 140 caregivers were recruited.

Table 1 displays the characteristics of the patients. The patients with non-oncological conditions are considerably older than those with oncological conditions (78.38  ±  9.98 years vs. 72.32  ±  11.90 years; *p*  =  0.001). They exhibit a statistically significant increased prevalence of comorbidities (*p*  =  0.04), with nearly half of them having more than four comorbidities. The majority of the patients with non-malignant conditions are entirely dependent on others for care or daily activities, as indicated by the Barthel score (96.83% compared to 58.44% of the patients with cancer; *p* < 0.00001). The time from diagnosis to the commencement of palliative treatment was markedly longer for the patients with non-malignant conditions (1098 days compared to 283 days; *p* = 0.001).

The response rate was 95.89%; three patient caregivers declined participation in the study, one patient died on the first day after admission, and two patients had no person involved in their care. The caregivers were divided into two groups: PrC_1_, which included the primary caregivers of patients with non-oncological illnesses (n = 63), and PrC_2_, which included the primary caregivers of patients with malignancies (n = 77). The demographic characteristics, including age, gender, living environment, degree of kinship, occupation, level of education, living place, the distance between the patient’s home and the caregiver, showed that there were no statistically significant differences between the two groups of primary caregivers (Table 2).

In half of the cases, the care of the patient was assumed by the subsequent generation, and in a quarter of the cases, the care was provided by their spouse. Almost 9 out of 10 patients were cared for by a family member and over two-thirds of these were women. Residence in the same locality or in a neighboring locality (expressed by a distance of less than 10 km) was identified in most cases. Around half of the caregivers were employed or were retired, with insignificant differences. The caregivers’ quality of life was measured by the MOS-SF36 and analyzed in two ways: the first way involved evaluating the dynamics from the initial median value to the last evaluated median value, and the second way involved comparing the median values between the oncological and non-oncological primary caregiver groups. Additionally, it was interesting to see the median value of the quality of life after a patient’s death (Tf). The median values of the eight parameters of quality of life are specified in Table 3.

All the aspects of quality of life assessed by the MOS-SF36 show a decrease in values as the patient approaches the end of their life. The caregiver is physically and emotionally exhausted.

The physical functionality of the caregivers decreased during the duration of care without statistical significance in either group, and it returned to the initial value 2 months after the patient’s death.

Bodily pain and emotional well-being are the two quality of life dimensions for which the statistical significance of the care process did not change two months after the patient’s death. However, for bodily pain and emotional well-being, it was observed that the initial value was half of the maximum value [19] and decreased without significance. Although there were no statistically significant differences between the two groups, the values obtained for those caring for patients with non-malignant diseases were lower.

In the first group (PrC1), the caregivers who cared for patients with non-malignant diseases had a decreased quality of life in terms of limitations in their usual role due to emotional problems (*p* = 0.002), social functioning (*p* = 0.01), energy/fatigue (*p* = 0.049), and their perception of their general health (*p* = 0.07) (Figure 2).

In the second group (PrC2), the caregivers of patients with a cancer diagnosis had impairments in the following aspects: limitations in their daily role due to physical health (*p* = 0.02), emotional problems (*p* = 0.05), and social functioning (*p* = 0.03) (Figure 3).

At the initial moment (T0), there were no significant differences between the two groups, except for limitations in the usual role due to emotional problems (*p* = 0.007) and the perception of general health (*p* = 0.02), which both had higher values in the group with oncological patients.

Physical functionality showed a slight but statistically insignificant decrease due to the increased patient needs throughout disease evolution. Bodily pain can be linked to the physical overload experienced by the carers.

In addition, limitations in the usual role due to physical health was a deeply affected aspect in both groups but recovered two months after the patients’ deaths, and was statistically significant in the group of cancer patients (*p* = 0.02).

Even if the change in the two aspects of quality of life was not statistically significant, it was observed that, especially in the case of bodily pain, it had a value less than half of the maximum value (45/100). As the disease progressed in the non-cancer patients, the body pain increased by 22.22% (35/100) due to the patient’s total dependence and the increase in their nursing needs. Two months after the patient’s death, the bodily pain improved by 27.78%, reaching an average value of 57.5/100 but remaining at just over half of the maximum value.

Both groups were affected by poor emotional well-being, which was moderately more severe in the caregivers who cared for patients with cancer, but without statistical significance. (48.3/100 in PrC1 versus 52.1/100 in PrC2; *p* = 0.26). These values remained approximately the same as the disease progressed due to the late evaluation. It was observed that two months after the patient’s death, the caregivers in both groups showed a slight improvement in their emotional status (15.89% in the non-oncological group and 15.18% in the oncological group), but this difference was statistically insignificant. The value to which the emotional well-being returned was only slightly above 50/100, which showed us that the caregiver remained at risk of developing certain emotional problems in the future. The influence of the role in patient care due to deficient emotional status was evident in both groups and was statistically significant (*p* = 0.007) in the group of those caring for patients with non-malignant diseases, probably due to the long duration of care. It was observed that the recovery of this aspect at the evaluation performed two months after the end of the care process was statistically significant in both groups (*p* = 0.002 in PrC1 and *p* = 0.05 in PrC2) but only in the proportion of two-thirds of the maximum value.

The vitality and health perception were markedly diminished in the PrC1 group; nevertheless, in both groups, the values remained below fifty percent of the maximum, which correlated directly with the physical and emotional conditions reported by the carers (*p* = 0.049). Regarding vitality, no statistically significant variations were seen between the two groups. The perception of overall health was more significantly impacted in the cohort caring for non-oncological patients at the initial assessment (*p* = 0.02). The social dimension indicates that the extensive responsibilities and heightened time dedicated to caregiving result in various social limitations, which did not exhibit statistically significant differences between the groups but were notably altered during the caregiving process, with *p* values of 0.01 in PrC1 and 0.03 in PrC2.

## 4. Discussion

In Romania, 76.5% of patients with palliative needs are cared for at home [22] and their care is provided by a family member in 88% of cases, similar to other studies in the literature that show a percentage of 85–90% [23,24]. On the one hand, most of the specialized palliative care services focus on cancer patients and few of them provide care for patients with non-malignant diseases [25]. Also, the distribution of specialized services is unequal in the country and many patients are cared for by their family [25]. The financial cost of palliative care in specialized units is more than four times that of home care [26,27] but the human cost of the care burden for the family and lost quality of life cannot be measured objectively [17,27]. Women are more frequently involved in caring for others, comprising two-thirds of caregivers. Gender is a factor that influences the burden of care: females are most vulnerable [27,28] because they have many responsibilities for the patient and other family members, and males have lower scores for anxiety and depression than females [17,29]. Among the primary caregivers of cancer patients, the quality of life was impacted in all dimensions, with statistically significant reductions and limitations in daily roles attributable to physical and emotional issues. This indicates that caregivers of cancer patients are often fatigued when accessing palliative care services. Despite this fact, caregivers of patients with non-malignant diseases experience a greater level of weariness [17].The most important predictive factor for primary caregivers’ quality of life is the burden of the caregiver, which is higher as the quality of their life decreases [30,31]. The second most important factor is the support from friends, other family members, and from the team of professionals, the latter being the one that can provide an improvement in the quality of life [32].

The situation may begin to return to normal after the patient’s death, but this does not seem to be fully achieved. Two months after the patient’s death (Tf), these parameters rise but do not exceed 60% of the normal value, indicating that the primary caregiver’s health is adversely affected by the caregiving process for their loved one. The emotional fluctuations in the caregiver’s mental state may evolve into psychiatric disorders, including depression and anxiety [29,33,34,35,36].

In the non-cancer cohort of primary caregivers, the circumstances are exacerbated by significant constraints in their typical roles, attributed to emotional and physical issues that indicate caregiver fatigue [17]. Moreover, the energy and vitality of caregivers diminish when attending to these patients [37]. The prolonged duration of caregiving may adversely impact the caregiver’s physical health, resulting in physiological discomfort and mental fatigue, accompanied by diminished energy levels.

The comparison between the oncological and non-oncological groups indicated a substantial decrease in the perception of general health regarding patients with non-malignant conditions [17]. The extensive duties and heightened time devoted to caregiving resulted in various social limitations that did not exhibit statistically significant differences across the groups but were notably altered during the caregiving process.

### Limitations

The sample used in this study was small. The main reason for this was patient death because the point when palliative care was initiated was late in their illness’ trajectory. This study did not evaluate the positive aspects of quality of life. The researchers did not analyze how the interventions of specialized teams influence the caregiver’s quality of life.

## 5. Conclusions

The large number of responsibilities, the long time spent caring for a patient, the uncertainty about the evolution of the disease, little knowledge about the disease, marginalization, and the lack of time for oneself are some of the elements that increase caregiver burden. Along with this, the quality of life of caregivers decreases significantly in different aspects, including physical, psycho-emotional, and social, with the perception of deteriorating general health. Physical and emotional exhaustion can be predictive factors for somatic or emotional pathology that caregivers may develop, even at an interval after the completion of the care process.

## Figures and Tables

**Figure 1 cancers-16-03570-f001:**
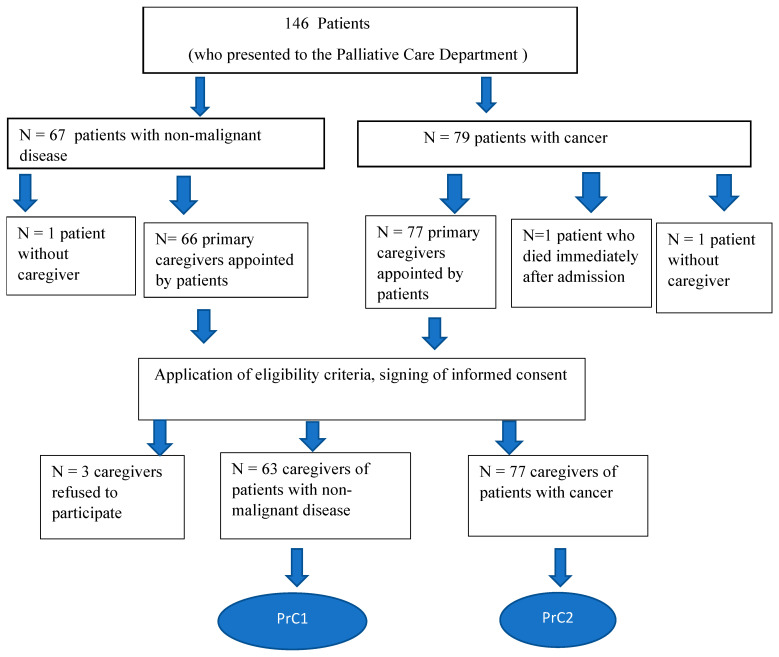
The algorithm for enrolling the subjects in the study.

**Figure 2 cancers-16-03570-f002:**
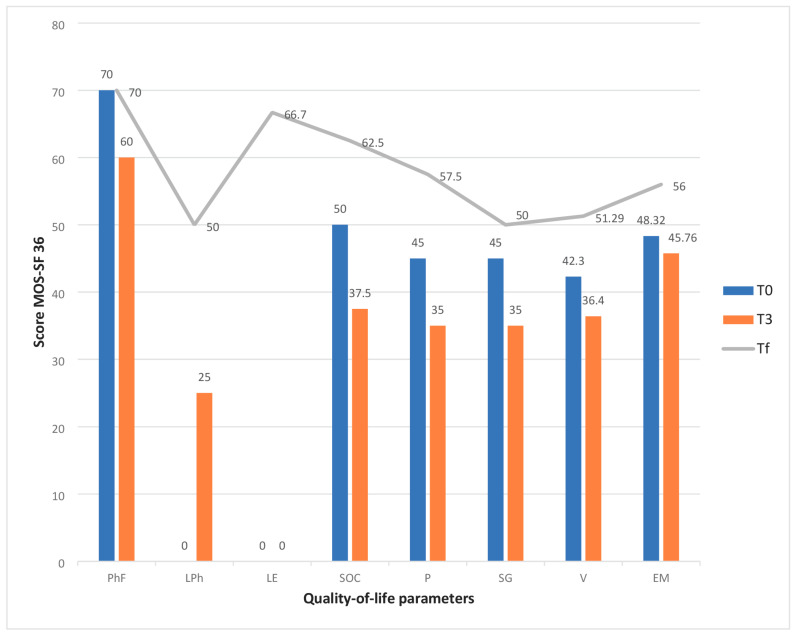
Changes in the quality-of-life of primary caregivers who care for non-oncological patients. Abreviations: PhF—Physical functioning, LPh—role limitations due to physical health, LE—role limitations due to emotional problems, SOC—social functioning, P—pain, GH—general health, V—vitality/energy, EM—emotional well-being, T0—time of inclusion in palliative care, T3—time of evaluation after three months, Tf—time of evaluation two months after the patient’s death.

**Figure 3 cancers-16-03570-f003:**
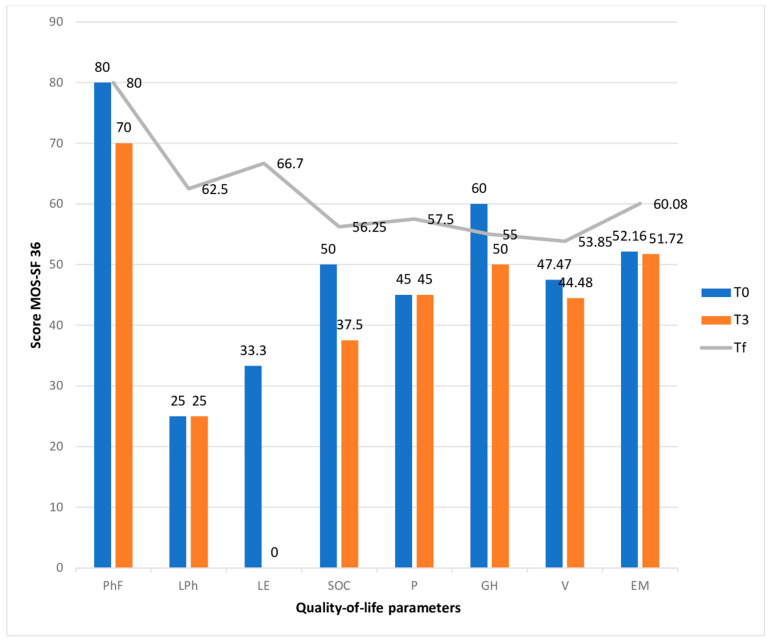
Changes in the quality-of-life of primary caregivers who care for cancer patients. Abreviations: PhF—Physical functioning, LPh—role limitations due to physical health, LE—role limitations due to emotional problems, SOC—social functioning, P—pain, GH—general health, V—vitality/energy, EM—emotional well-being, T0—time of inclusion in palliative care, T3—time of evaluation after three months, Tf—time of evaluation two months after the patient’s death.

**Table 1 cancers-16-03570-t001:** Characteristics of patients included in study (P1—group of patients with non-oncological diseases and P2—group of patients with cancer).

Parameter	Category	Non-Oncological Group (P1)	Oncological Group (P2)	*p*
Age (mean ± SD)	Years	78.38 ± 9.981	72.32 ± 11.909	**0.001**
Gender	Male n (%)	42 (66.67%)	52 (67.53%	0.91
	Female n(%)	21 (33.33%)	25 (32.47%)	
Living environment	Rural n(%)	30 (47.62%)	28 (36.36%)	0.17
	Urban n(%)	33 (52.38%)	49 (63.64%)	
Occupation	Employee n(%)	1 (1.59%)	3 (3.90%)	0.65
	Retired n(%)	62 (98.41%)	74 (96.10%)	
Comorbidities	1 n(%)	0	9 (11.68%)	**0.02**
	2–4 chronic diseases n(%)	36 (57.14%)	42 (54.55%)	0.34
	≥5 chronic diseases n(%)	27 (42.86%)	26(33.77%)	**0.04**
Barthel Score	100–59 partially dependent n(%)	2 (3.17%)	21 (27.27%)	**0.0001**
	20–39 very dependent n(%)	2 (3.17%)	11 (14.29%)	**0.024**
	<20 totally dependent n(%)	59 (93.66%)	45 (58.44%)	**<0.00001**
The time from diagnosis to palliative care	Median (days)	1098	283	**0.001**
		(82–2747)	(69–761)	
Days of hospital admission in the last six months	Median (days)	14 (8–23)	14 (5–25)	0.30

**Table 2 cancers-16-03570-t002:** Demographic characteristics of the two groups.

Parameter	Category	Non-Oncological Group (PrC1)	Oncological Group (PrC2)	*p* Value
Age (years)	Mean ± SD	58.32 ± 12.417	54.76 ± 12.52	0.06
Gender	Male N(%)	20 (31.74)	23 (29.87)	0.81
	Female N(%)	43 (68.26)	54 (70.13)	0.81
Living environment	Rural N(%)	17 (26.99)	21 (27.27)	0.96
	Urban N(%)	46 (73.01)	56 (72.73)	0.96
Degree of kinship	Husband/wife N(%)	15 (23.8)	20 (25.97)	0.83
	Brother/sister N(%)	2 (3.18)	3 (3.90)	0.92
	Son/daughter N(%)	34 (53.97)	40 (51.95)	0.82
	Nephew/niece N(%)	4 (6.35)	5 (6.50)	0.91
	Others N(%)	8 (12.7)	9 (11.68)	0.81
Occupation	Employee N(%)	22 (34.92)	38 (49.35)	0.83
	Retired N(%)	32 (50.8)	35 (45.45)	0.52
	Unemployed N(%)	9 (14.28)	4 (5.20)	0.06
Level of education	Primary education N(%)	2 (3.18)	4 (5.20)	0.3
	Gymnasium studies N(%)	9 (14.28)	15 (19.48)	0.41
	High school (college) N(%)	36 (57.14)	31 (40.25)	**0.04**
	Higher education N(%)	16 (25.40)	27 (35.05)	0.21
Living place	Similar to that of the patient N(%)	40 (63.49)	38 (49.35)	0.93
	Different from the patient’s N(%)	23 (36.51)	39 (50.65)	0.93
Distance between the patient’s home and the caregiver	Under 1 km N(%)	41 (65.08)	46 (59.74)	0.51
	1–10 km N(%)	12 (19.05)	16 (20.78)	0.79
	10–30 km N(%)	4 (6.35)	6 (7.80)	0.74
	Over 30 km	6 (9.52)	9 (11.68)	0.68

**Table 3 cancers-16-03570-t003:** The parameters of caregiver’s quality of life during patient care (Abbreviations: PhF—physical functioning, LPh—role limitations due to physical health, LE—role limitations due to emotional problems, SOC—social functioning, P—pain, GH—general health, V—vitality/energy, EM—emotional well-being, T0—time of inclusion in palliative care, T3—time of evaluation after three months, Tf—time of evaluation two months after the patient’s death).

Parameter		Non-Malignant Group (PrC1)	Oncologic Group (PrC2)	*p* Value (Test Mann–Whitney)
Physical functionality (PhF)	Initial value (T0)	70 (55, 95)	80 (47.5, 90)	0.86
	The last value evaluated (T3)	60 (35, 80)	70 (42.5, 92.5)	0.6
	Final value (Tf)	70 (55, 95)	80 (60, 93.75)	0.77
	***p* value (Kruskal–Wallis test)**	0.3	0.53	-
Limitations in usual role due to physical health (LPh)	Initial value (T0)	0 (0, 50)	25 (0, 75)	0.13
	The last value evaluated (T3)	25 (0, 50)	25 (0, 75)	0.47
	Final value (Tf)	50 (0, 100)	62.5 (25, 100)	0.23
	***p* value (Kruskal–Wallis test)**	0.13	**0.02**	-
Limitations in usual role due to emotional problems (LE)	Initial value (T0)	0 (0, 33)	33.3 (0, 100)	**0.007**
	The last value evaluated (T3)	0 (0, 33)	0 (0, 66.7)	0.29
	Final value (Tf)	66.7 (0, 100)	66.7 (0, 100)	0.55
	***p* value (Kruskal–Wallis test)**	**0.002**	**0.05**	-
Social functioning (SOC)	Initial value (T0)	50 (25, 62.5)	50 (25, 75)	0.32
	The last value evaluated (T3)	37.5 (25, 50)	37.5 (25, 68.75)	0.26
	Final value (Tf)	62.5 (37.5, 75)	56.25 (37.5, 75)	0.62
	***p* value (Kruskal–Wallis test)**	**0.01**	**0.03**	-
Bodily pain (P)	Initial value (T0)	45 (22.5, 57.5)	45 (45, 75)	0.09
	The last value evaluated (T3)	35 (22.5, 57.5)	45 (22.5, 57.5)	0.85
	Final value (Tf)	57.5 (22.5, 90)	57.5 (45, 77.5)	0.95
	***p* value (Kruskal–Wallis test)**	0.13	0.06	-
General health (GH)	Initial value (T0)	45 (30, 60)	60 (40, 70)	**0.02**
	The last value evaluated (T3)	35 (15, 55)	50 (25, 62.5)	0.07
	Final value (Tf)	50 (30, 70)	55 (40, 75)	0.16
	***p* value (Kruskal–Wallis test)**	0.07	0.17	-
Vitality (energy/fatigue) (V)	Initial value (T0)	42.3 (0, 90)	47.47 (0, 100)	0.19
	The last value evaluated (T3)	36.4 (0, 85)	44.48 (20, 100)	0.13
	Final value (Tf)	51.29 (5, 95)	53.85 (10, 90)	0.63
	***p* value (Kruskal–Wallis test)**	**0.049**	0.12	
Emotional well-being (EM)	Initial value (T0)	48.32 (8, 96)	52.16 (8, 100)	0.26
	The last value evaluated (T3)	45.76 (0, 84)	51.72 (8, 96)	0.32
	Final value (Tf)	56 (8, 92)	60.08 (16, 100)	0.42
	***p* value (Kruskal–Wallis test)**	0.13	0.08	

## Data Availability

The original contributions presented in the study are included in the article, further inquiries can be directed to the corresponding author.

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
