# Peer review of "Decrease in Primary Caregivers’ Quality of Life During the Care of a Relative with Palliative Care Needs: A Prospective Longitudinal Study"

_cancers, 2024, doi:10.3390/cancers16213570_

Round 1
Reviewer 1 Report
Comments and Suggestions for Authors
Decrease in primary caregiver’s quality of life during the care 2 of a relative with palliative care needs: A prospective longitudinal study
This is a very interesting manuscript and I read it with much appreciation. I believe it will add valuably to the knowledge about palliative care caregivers.
Comments are found below.
Line 25 - decrease in their usual role – what role do you mean?
Figure 1 is incomplete N=77 primary caregivers appointed by…?
Line 99 – check the reference form
Line 104 – Kruskal-Wallis H test: were the results skewed or normally distributed?
Line 113-114 reverse the groups, as the PrC1 is the first mentioned everywhere else in the manuscript
Table 1 – either use the % [N=63 (%) and N=77 (%)] after the age line or remove it from the 1st line and put it on each subsequent line. Also, was age normally distributed?
Table 2 – please explain the numbers inside the parenthesis and explain the Kruskal Wallis result: since not all patients died, is the difference between T0 and T3 different for all values?
Also, please provide with numbers on text or table or figure of how many people on palliative care died within the 3-month interval of the study in each group. Also, provide a clear p value between your provided values on each group and in between them on T0 and T3 on patients that survived the 3 month mark.
Line 147 – provide reference for “normal value”
Line 161-162 – group PrC2 does not include any oncological patients
Line 164 - In parallel, bodily pain occurs due to physical overload – was this a question in the questionnaire?
Line 176 – again, there are no oncology groups in the study. Also, you have an extra full-stop before the parenthesis.
Line 178 - due to the late evaluation when a patient had already adapted to the tense situation – please explain better how this is shown in the results
Line 193 - directly related – please provide with a correlation value
Line 211- women have not been proven to have “natural skills” in housekeeping, caring for the body, and preparing food, nor does the reference provided state such a comment (ref 28). Such patriarchal statements help zero to women, who do hold the majority of the burden of palliative care. Also, ref 28 is a beautiful article about family and spirituality – however, it is (as clearly stated inside the article) not diverse in culture appreciation and should not be used to support the Authors’ patriarchal claims.
Line 217-218 – “This shows that cancer patient’s caregivers are usually exhausted when accessing a palliative care service, especially in cases where they care for patients with non-malignant diseases [17].“ Please rephrase because it seems like this specific group of caregivers care for cancer AND noncancer patients and have nothing to do with your study.
Line 220 – “which is higher as the quality of his life decreases [31]” – please use a more appropriate reference as schizophrenia quality of life cannot possibly be compared to the quality of life from cancer or non-oncological palliative care needing disease. It is a disservice to both mental illness and palliative care patients. May I suggest “PMC9892678/” and “PMC4289985/” if you are looking for non-palliative alternates.
Reference 32 – As stated above, mental illness is many times incomparable to palliative care patients. I propose (although you might find better articles) https://pubmed.ncbi.nlm.nih.gov/33242701/ and PMC7975852, and PMC7796575/
Line 226 – define normal values and provide references
Comments on the Quality of English Language
Line 15 and line 16- his or their?
Line 20 – contained, do you mean included?
Line 50 – inconstant – do you mean inconstantly?
Line 54 – evaluation of, and impact on
Line 84 – at the initial moment
Line 93 – instrument, do you mean tool?
Line 100 – instrument or tool? Also, has a good internal consistency, and check the reference form
Line 123 – domicile, do you mean residence?
Line 125 – status of employment, do you mean employed? Also, why are employed used in a past tense and retired in a present tense? Did this change throughout the study?
Line 134 - LPh- rol limiptations – please edit
Line 135 - V- vitalitt/energy – please edit
Line 137 – monts – please edit
Figure 2 and Figure 3 – please edit the title and under title of the figure (limiptations, vitalitt, monts etc)
Reviewer 2 Report
Comments and Suggestions for Authors
Cancers
Decrease in primary caregiver’s (NOTE—should be “caregivers’) quality of life during the care of a relative with palliative care needs.
This very well-written paper seeks to assess caregivers’ quality-of-life over time as they
helped patients in palliative care who were or were not diagnosed with cancer. This implies that differences in the needs of patients with and without cancer diagnoses differentially impacted caregivers. However, no data are presented on patient demand so no conclusions can be drawn here. In addition, authors conclude that “the large number of responsibilities, the long time spent caring for a patient and uncertainty about the evolution of the disease, little knowledge about the disease, marginalization, and lack of time for oneself are some of the elements that increase caregiver burden.” This implies that these elements were measured, but unfortunately, they were not. The absence of data on patient demand and specifics of these demands on caregiver well-being minimizes the value of these findings.
The principle finding, therefore is demonstration of a modest deterioration of caregiver well-being over time, with some improvement following the death of the patients. This adds little to the rapidly expanding data on the adverse impact of caring for declining critically ill patients which, as the authors attest, falls mainly on women who are members of patients’ families.
The major data in this study were collected via the Rand Corporation’s 36-item Short form Survey Instrument. Since this research depends heavily on data produced by use of this scale, it would be useful discuss its validity a measure of caregiver stress and why it was selected instead of Rand’s 20-item version of the same scale or other more widely used measures of quality of life, e.g. the 18-itemGeneral Well-being Schedule produced by the US National Institute of Health (https://cde.nlm.nih.gov/formView?tinyId=YkER84OKU).
With discussion of the points above, and combining data from both patient groups, these data could warrant a brief publication documenting changes in caregiver well-being over time, with literature-based hypotheses about possible explanatory factors.
This could facilitate further research in this very important area.
As stated, the authors write brilliantly in a language not their own, they are working in an area that has great worldwide importance, and they have access to a valuable source of data. I look forward to seeing their future contributions.
Round 2
Reviewer 1 Report
Comments and Suggestions for Authors
Thank you for addressing my comments, good work!
Comments on the Quality of English LanguagePlease have a native speaker proof your manucript, as I am not one.